# Reproducibility and Accuracy of the Radiofrequency Echographic Multi-Spectrometry for Femoral Mineral Density Estimation and Discriminative Power of the Femoral Fragility Score in Patients with Primary and Disuse-Related Osteoporosis

**DOI:** 10.3390/jcm11133761

**Published:** 2022-06-29

**Authors:** Piera Lalli, Claudia Mautino, Chiara Busso, Francesca Bardesono, Marco Di Monaco, Lorenzo Lippi, Marco Invernizzi, Marco Alessandro Minetto

**Affiliations:** 1Division of Physical Medicine and Rehabilitation, Department of Surgical Sciences, University of Turin, 10126 Turin, Italy; plalli@cittadellasalute.to.it (P.L.); chiara.busso@unito.it (C.B.); 2Division of Spinal Unit, Department of Orthopedics, Traumatology and Rehabilitation, “Città Della Salute e Della Scienza” University Hospital, 10126 Turin, Italy; cmautino@cittadellasalute.to.it; 3Division of Physical and Rehabilitation Medicine, Osteoporosis Research Center, Presidio Sanitario San Camillo, Opera San Camillo Foundation, 10131 Turin, Italy; francesca.bardesono@gmail.com (F.B.); marco.di.monaco@alice.it (M.D.M.); 4Department of Health Sciences, University of Eastern Piedmont “A. Avogadro”, 28100 Novara, Italy; lorenzolippi.mt@gmail.com (L.L.); marco.invernizzi@med.uniupo.it (M.I.); 5Dipartimento Attività Integrate Ricerca e Innovazione (DAIRI), Translational Medicine, Azienda Ospedaliera SS. Antonio e Biagio e Cesare Arrigo, 15121 Alessandria, Italy

**Keywords:** bone mineral density, fracture risk, fragility score, FRAX, spinal cord injury

## Abstract

We aimed to investigate the reproducibility and accuracy of Radiofrequency Echographic Multi-Spectrometry (REMS) for femoral BMD estimation and the reproducibility and discriminative power of the REMS-derived femoral fragility score. 175 patients with primary and disuse-related osteoporosis were recruited: one femoral Dual-energy X-ray Absorptiometry (DXA) scan and two femoral REMS scans were acquired. No significant test—retest differences were observed for all REMS-derived variables. The diagnostic concordance between DXA and REMS was 63% (Cohen’s kappa = 0.31) in patients with primary osteoporosis and 13% (Cohen’s kappa: −0.04) in patients with disuse-related osteoporosis. No significant difference was observed between REMS and DXA for either femoral neck BMD (mean difference between REMS and DXA: −0.015 g/cm^2^) or total femur BMD (mean difference: −0.004 g/cm^2^) in patients with primary osteoporosis. Significant differences between the two techniques were observed in patients with disuse-related osteoporosis (femoral neck BMD difference: 0.136 g/cm^2^; total femur BMD difference: 0.236 g/cm^2^). Statistically significant differences in the fragility score were obtained between the fractured and non-fractured patients for both populations. In conclusion, REMS showed excellent test-retest reproducibility, but the diagnostic concordance between DXA and REMS was between minimal and poor. Further studies are required to improve the REMS—derived estimation of femoral BMD.

## 1. Introduction

Dual-energy X-ray Absorptiometry (DXA) is widely accepted as the gold standard technique for the assessment of bone mineral density (BMD) at the lumbar spine, proximal femur, and distal radius. This assessment is the basis for osteoporosis diagnostic classification and fracture risk estimation [1]. In recent years, an ultrasound–based technique has also been developed to estimate BMD for the lumbar spine and femoral neck, with the name of Radiofrequency Echographic Multi-Spectrometry (REMS) [2]. Its operating principle is the analysis of the radiofrequency signals acquired during an ultrasound scan of the lumbar vertebrae and of the femoral neck. Ultrasound signals are processed to derive a patient-specific spectrum that is compared with gender-, age-, site-, and body mass index-matched reference spectral models to estimate the BMD and the fragility score [2,3]. Recent studies showed high accuracy and precision of the REMS approach in comparison with DXA for the diagnosis of osteoporosis on the lumbar spine and femoral neck in postmenopausal women [3,4,5]. Adami et al. [6] have also showed that the REMS-derived T-score was an effective predictor for the risk of fragility fracture in postmenopausal women. Other very recent studies showed the feasibility and usefulness of REMS in pregnant women [7], in elderly women with type 2 diabetes [8], and in acromegaly patients [9]. To our knowledge, no previous study has investigated the accuracy of REMS in comparison with DXA in the secondary (disuse-related) osteoporosis occurring in spinal cord injured patients. The bone loss in these patients is always associated with disuse-related changes of muscle size and quality [10,11,12,13] and can also be accompanied by the presence of heterotopic bone formations known as para-osteoarthropathies (i.e., periarticular ossifications) that frequently involve the soft tissues around the hip [10,11,14]. However, the changes of muscle size and quality as well as the presence of para-osteoarthropathies could affect the radiofrequency signals acquired during an ultrasound scan of the proximal femur [13,14]. Therefore, we hypothesized that the accuracy of REMS for the diagnosis of osteoporosis on the femoral neck could be limited in patients with disuse-related osteoporosis.

The REMS-derived fragility score (i.e., a dimensionless number in the range 0–100) has been proposed as a BMD-independent indicator of bone quality: the lower the value of the fragility score, the higher the quality of the considered bone target [2]. Preliminary studies showed that the lumbar spine fragility score discriminates between fractured and non-fractured patients [15] and correlates with the FRAX-derived 10-year probabilities of major and hip fracture [16]. To our knowledge, no previous study has investigated the reproducibility and discriminative power of the femoral fragility score in patients with osteoporosis.

Therefore, the aim of this study was to fill the above-mentioned research gaps (no previous study adopted REMS to investigate spinal cord injured patients and no previous study investigated the reproducibility and discriminative power of the femoral fragility score in osteoporotic patients) by analyzing the reproducibility and accuracy of REMS for femoral BMD estimation and the reproducibility and discriminative power of the femoral fragility score in patients with primary and disuse-related osteoporosis.

## 2. Materials and Methods

### 2.1. Study Design and Participants

This was a cross-sectional study performed in patients of both genders who were recruited from the rehabilitation units of five different hospitals: AOU Città della Salute e della Scienza (Torino, Italy), Presidio Sanitario San Camillo (Torino, Italy), Casa di Cura Villa Serena (Piossasco, Italy), Presidio Riabilitativo Borsalino (Alessandria, Italy), Istituto Clinico Scientifico Maugeri (Torino, Italy). Inclusion criteria were: age ≥18 years and medical prescription for femoral DXA. The exclusion criteria were body mass index ≥ 40 kg/m^2^ and presence of osteoporosis secondary to disorders other than the spinal cord injury: other causes of secondary osteoporosis were excluded on the basis of anamnestic, clinical, and biochemical findings.

The American Spinal Injury Association (ASIA) impairment scale [17] was adopted to assess the disease severity in patients with spinal cord injury.

A total sample of 175 patients (primary osteoporosis: *n* = 140; disuse-related osteoporosis: *n* = 35) was recruited. All participants provided their written informed consent before participating.

Ethics approval (protocol n. 133282) was granted by Ethics Committee of the University of Turin (Italy) and the procedures were conducted according to the Declaration of Helsinki.

### 2.2. DXA and REMS

One femoral DXA scan and two femoral REMS scans (of the same side investigated by DXA) were acquired in all patients: the median interval between the DXA and REMS acquisitions was 29 days.

DXA scanning of the proximal femur (the left side was investigated in all non–fractured patients, while the non–fractured side was investigated in all patients with a previous femoral fracture) was performed using Hologic scanner (models: Horizon A, Horizon Wi, Discovery Wi-Hologic Inc., Bedford, MA, USA) or GE scanner (models: Lunar Prodigy, Lunar iDXA-GE-Lunar Inc., Madison, WI, USA) according to the standard clinical routine procedures. Since the BMD_REMS_ was highly correlated with BMD_DXA_ measurements obtained with Hologic densitometers [18,19], the comparison between BMD_DXA_ and BMD_REMS_ was performed after the conversion of BMD_DXA_ values obtained by GE scanners in Hologic–equivalent values, as previously described [3], by using the following equations: femoral neck standardized BMD (g/cm^2^) = ((0.939 × observed BMD) − 0.023] [20]; total femur standardized BMD (g/cm^2^) = [(0.979 × observed BMD) − 0.031) [21]. For the purposes of the present study, diagnostic classifications (osteopenia or osteoporosis) were based on the T-score values.

Two REMS scans of the proximal femur (of the same side investigated by DXA) were performed by the same physician (MAM) with 10 years of experience in musculoskeletal ultrasonography: after the first scan, the subject was allowed to move and the transducer was repositioned. The REMS scans were performed with a dedicated echographic device (EchoStation, Echolight SpA, Lecce, Italy) equipped with a convex transducer that was placed parallel to the head-neck axis in order to visualize the proximal femur profile. For the two acquisitions performed in each patient, the transducer focus and scan depth were adjusted to have the target bone interface in the ultrasound beam focal zone at about halfway through the B-mode image depth. REMS acquisitions were analyzed with the EchoStudio (version 2.0-Echolight SpA, Lecce, Italy) software package.

A rigorous quality check of all examinations was performed a posteriori in order to guarantee the maximum reliability of the diagnostic output [3,4]. Two experienced operators (PL, CB) checked all DXA and REMS reports in order to identify possible acquisitions errors (see Section 3).

### 2.3. Statistical Analysis

Changes in the values of femoral neck BMD, total femur BMD, and fragility score between the two REMS scans (REMS 1 vs. REMS 2) were analyzed with the Wilcoxon signed rank sum test to assess the presence of systematic bias and with the intraclass correlation coefficient (ICC2,1: two–way mixed model, single measures, absolute agreement) to assess the REMS reproducibility. A sample size of at least 30 subjects (in each of the two groups) was considered necessary for the test-retest reproducibility analysis, using the approximate method developed by Walter et al. [22] based on α = 0.05 and β = 0.20, indicating an expected level of reproducibility (ρ_1_) of 0.98 [3] and a minimally acceptable level of reproducibility (ρ_0_) of 0.95.

The criteria used for the interpretation of the ICCs were as follows: 0.00–0.25: no correlation; 0.26–0.49: low correlation; 0.50–0.69: moderate correlation; 0.70–0.89: high correlation; and 0.90–1.00: very high correlation [23]. The ICC was also adopted to estimate the standard error of measurement (SEM: standard deviation of all values obtained with the two scans × √1-ICC) and the smallest detectable change (SDC = SEM × 1.96 × √2) [24]. Moreover, the least significant change (LSC) was also calculated for the three variables by using the ISCD Precision Calculating Tool (https://iscd.org/learn/resources/calculators/, accessed on 14 February 2022).

The diagnostic concordance between DXA end REMS was assessed by calculating the diagnostic agreement percent (i.e., the percentage of patients being classified in the same diagnostic category) and Cohen’s kappa. The criteria used for the interpretation of the Cohen’s kappa were as follows: negative values and positive values in the range 0–0.20: no agreement; 0.21–0.39: minimal agreement; 0.40–0.59: weak agreement; 0.60–0.79: moderate agreement; 0.80–0.90: strong agreement; >0.90: almost perfect agreement [25].

The differences between the values of femoral neck BMD and total femur BMD obtained by DXA and REMS were analyzed with the Wilcoxon signed rank sum test. Additionally, the Bland–Altman method was adopted to show the differences between the two measurements against their averages and to check whether the measurement difference was independent of the magnitude of the average: the strength of the relationship between differences and averages was evaluated by the Spearman test.

The Mann–Whitney U test was adopted to compare the fragility scores between different subgroups (fractured vs. non-fractured patients) and the Spearman test was adopted to investigate the association between fragility score and the FRAX-derived 10-year probabilities of fracture [26,27].

Data were expressed as median and 1st–3rd quartile and threshold for statistical significance was set to *p* = 0.05. All statistical tests were performed with the IBM SPSS Statistics (version 20-IBM Corporation, Armonk, NY, USA) software package, with the exception of the Bland-Altman plots that were performed with MedCalc (version 19-MedCalc Software Ltd., Ostend, Belgium) software package.

## 3. Results

The clinical characteristics of two groups of patients are reported in Table 1.

After exclusion of 25 and five erroneous DXA reports (due to inaccurate patient positioning and/or incorrect placement of the regions of interest in the image) in patients with primary and disuse-related osteoporosis, respectively, and after exclusion of five and four erroneous REMS scans (due to wrong or suboptimal settings of the transducer position or focus and/or scan depth) in patients with primary and disuse-related osteoporosis, 135 patients with primary osteoporosis and 31 patients with disuse-related osteoporosis were considered for REMS reproducibility assessment, while 113 patients with primary osteoporosis and 30 patients with disuse-related osteoporosis were considered for REMS accuracy assessment.

Figure 1 shows the comparisons of femoral neck BMD, total femur BMD, and fragility score between the two REMS acquisitions (left panels: primary osteoporosis; right panels: disuse-related osteoporosis).

No significant test—retest differences were observed in both groups of patients (*p* > 0.05 for all comparisons). The ICC—SEM—SDC—LSC values obtained from the comparisons between the two REMS acquisitions in patients with primary and disuse-related osteoporosis are reported in Table 2. Given the excellent (ICC values > 0.90) [23] reproducibility of REMS, data from the two REMS acquisitions were averaged to perform the subsequent analyses (REMS accuracy assessment and fragility score comparisons between different subgroups).

The diagnostic concordance between DXA and REMS was 63% (Cohen’s kappa = 0.31, *p* = 0.0001) in patients with primary osteoporosis and 13% (Cohen’s kappa: −0.04, *p* = 0.512) in patients with disuse-related osteoporosis.

Figure 2 shows the comparisons of femoral neck BMD and total femur BMD between DXA and REMS and the relative Bland-Altman plots in patients with primary osteoporosis: no significant difference was observed between the two techniques for either femoral neck BMD (mean difference between REMS and DXA of −0.015 g/cm^2^) or total femur BMD (mean difference of −0.004 g/cm^2^). No significant correlations were observed between the differences and the averages for either femoral neck BMD (R = −0.065, *p* = 0.49) or total femur BMD (R = −0.013, *p* = 0.88).

Figure 3 shows the comparisons of femoral neck BMD and total femur between DXA and REMS and the relative Bland-Altman plots in patients with disuse-related osteoporosis: significant differences were observed between the two techniques for both femoral neck BMD (mean difference between REMS and DXA of 0.136 g/cm^2^) and total femur BMD (mean difference of 0.236 g/cm^2^). The Spearman test showed significant negative correlations between the differences and the averages for both femoral neck BMD (R = −0.36, *p* = 0.05) and total femur BMD (R = −0.55, *p* = 0.001), thus indicating that the lower the BMD values, the higher the REMS overestimation.

Figure 4 shows the comparisons of fragility score values between non-fractured and fractured patients with primary and disuse-related osteoporosis: statistically significant differences were obtained between the two subgroups of patients in both populations. Significant positive correlations were observed between fragility score and FRAX-derived 10-year probability of both major fractures (R = 0.65, *p* = 0.0001) and hip fracture (R = 0.62, *p* = 0.0001) in the subgroup of non-fractured patients with primary osteoporosis.

## 4. Discussion

This study investigated the reproducibility and accuracy of REMS for femoral BMD estimation and the reproducibility and discriminative power of the REMS-derived femoral fragility score in patients with primary and disuse-related osteoporosis. The main findings of our study were: (i) REMS showed excellent test-retest reproducibility for the estimation of femoral neck BMD, total femur BMD, and femoral fragility score in both populations of patients; (ii) no significant differences (i.e., no systematic bias) were observed between DXA and REMS for the estimation of femoral neck BMD and total femur BMD in patients with primary osteoporosis. However, the diagnostic concordance between the two techniques was minimal; (iii) REMS overestimated (with respect to DXA) both femoral neck BMD and total femur BMD in patients with disuse-related osteoporosis: the lower the BMD values, the higher the REMS overestimation. Moreover, the diagnostic concordance between the two techniques was poor; (iv) REMS–derived femoral fragility score differed between non-fractured and fractured patients in both populations of patients.

The demonstration of excellent test-retest reproducibility of REMS was obtained in a large group of patients of both genders with primary and disuse-related osteoporosis: this finding confirms and extends previous results that were obtained for both lumbar spine BMD and femoral neck BMD in small (*n* = 30) groups of women with postmenopausal osteoporosis [3,4,5]. The SDC and LSC values we obtained for both femoral neck BMD and total femur BMD are within the range of the measurement precision recommended for the DXA-derived estimations of both femoral neck BMD (LSC ≤ 6.9%) and total femur BMD (LSC ≤ 5%) [28]. However, further studies are required to assess the long-term reproducibility of the REMS estimates in different populations of patients with primary and secondary osteoporosis.

Previous studies performed in large groups of postmenopausal women showed moderate to strong diagnostic agreement between DXA and REMS. In fact, Di Paola et al. found a diagnostic concordance for femoral neck of 88% (Cohen’s kappa = 0.79) [3], while Cortet et al. [4] showed an age-dependent reduction of diagnostic concordance and Cohen’s kappa that were (for femoral neck) 92.1% and 0.90, respectively, in women aged 30–50 years, 84.9% and 0.84 in women aged 51–70 years, and 83.9% and 0.75 in women aged 71–90 years. Recent real-life experience studies performed in either postmenopausal women [5] or subjects of both genders with primary osteoporosis [29] showed weak to moderate diagnostic agreement between DXA and REMS: diagnostic concordance and Cohen’s kappa (for femoral neck) were 71.4% and 0.53 in the former study [5] and 84.8% and 0.67 in the latter study [29]. We found minimal (in patients with primary osteoporosis) and poor (in patients with disuse-related osteoporosis) diagnostic agreement between DXA and REMS. All previous above-mentioned studies investigated adult and elderly non-fractured women (with the exception of the study by Nowakowska-Plaza et al. who studied subjects of both genders) [29], while our sample of patients with primary osteoporosis consisted in non-fractured and fractured elderly patients of both genders. Therefore, the discrepancy between the minimal diagnostic agreement between DXA and REMS we obtained in our sample of patients with primary osteoporosis and the weak to strong agreement observed in previous studies can be related to inter-sample differences in age, gender, and disease severity. Moreover, an additional factor could underlie the poor diagnostic agreement between DXA and REMS and the REMS overestimation of femoral BMD we obtained in our sample of patients with disuse-related osteoporosis. Although no direct measurements of fat and muscle size and quality were obtained in our study, we hypothesize that the poor accuracy of REMS in patients with disuse-related osteoporosis can be related to the well-known (although not completely understood) effects of disuse on subcutaneous fat size, muscle size, and muscle composition [13,14]. The increased subcutaneous fat thickness as well as the atrophy and myosteatosis of the iliopsoas muscle can modify the ultrasound propagation between the ultrasound probe and the femoral neck. Consistently, it has been documented in different neuromuscular disorders that the replacement of muscle tissue by fat and fibrous tissue results in many transitions between tissues with different acoustic impedance [30,31,32]: these transitions produce reflections and attenuation of the ultrasound beam that could therefore modify the ultrasound signal spectrum and ultimately bias the BMD estimation. However, a limitation of our study is that the number of patients with disuse-related osteoporosis we included was relatively small: thus, significant differences between DXA and REMS could also be a result of type I error. Future studies are therefore required to confirm our preliminary observation as well as to improve the ultrasound signal processing and the REMS—derived estimation of femoral BMD in patients with spinal cord injury. This population of patients would indeed benefit from the availability of an easy-to-use, portable, and radiation-free approach that can be adopted for early diagnosis in the first weeks after the injury (when the execution of a DXA investigation is not easily feasible) and for short- and long-term follow-up.

Spectral analyses of ultrasound signals can be used not only for estimating the BMD of the bone target, but also for obtaining the fragility score that has been proposed as a BMD-independent marker of bone quality [2,15,16]. The present demonstrations of excellent reproducibility and between-subgroup (non-fractured vs. fractured patients) differences in femoral fragility score represent original results of our study, while the observed positive correlations between fragility score and FRAX-derived estimates of fracture probabilities extend previous findings that were obtained for the lumbar spine fragility score [15,16]. In the present series, we also observed that several non-fractured patients with primary osteoporosis had a fragility score above the median value of fragility score (57%: Figure 4a) obtained in the population of fractured patients: this subgroup of non-fractured patients could probably be considered at increased fracture risk. However, further studies are required to establish whether the fragility score has true discriminative power for different subgroups of patients (e.g., its discriminative power could be proved through a longitudinal study investigating the occurrence of fragility fractures in non-fractured patients presenting different baseline fragility scores) and can therefore be considered a relevant predictor of the fracture risk similar to other well-known BMD-independent predictors such as advancing age, previous fracture, glucocorticoid therapy, family history of hip fracture, current smoking, and trabecular bone score (TBS) [33,34]. The latter is a texture index obtained from the lumbar spine DXA that can be used to improve the FRAX-derived estimates of fracture probabilities [34,35]. A possible implication of our findings is that the fragility score could also be used to adjust the FRAX-derived estimates if further studies will confirm its fracture risk predictive value.

## 5. Conclusions

REMS showed excellent test-retest reproducibility for the estimation of femoral neck BMD, total femur BMD, and femoral fragility score in patients with primary and disuse-related osteoporosis. Moreover, the REMS–derived femoral fragility score differed between non-fractured and fractured patients in both populations of patients. However, the diagnostic concordance between DXA and REMS was minimal in patients with primary osteoporosis and poor in patients with disuse-related osteoporosis. Further studies are required to improve the ultrasound signal processing and the REMS—derived estimation of femoral BMD, especially in patients with spinal cord injury.

## Figures and Tables

**Figure 1 jcm-11-03761-f001:**
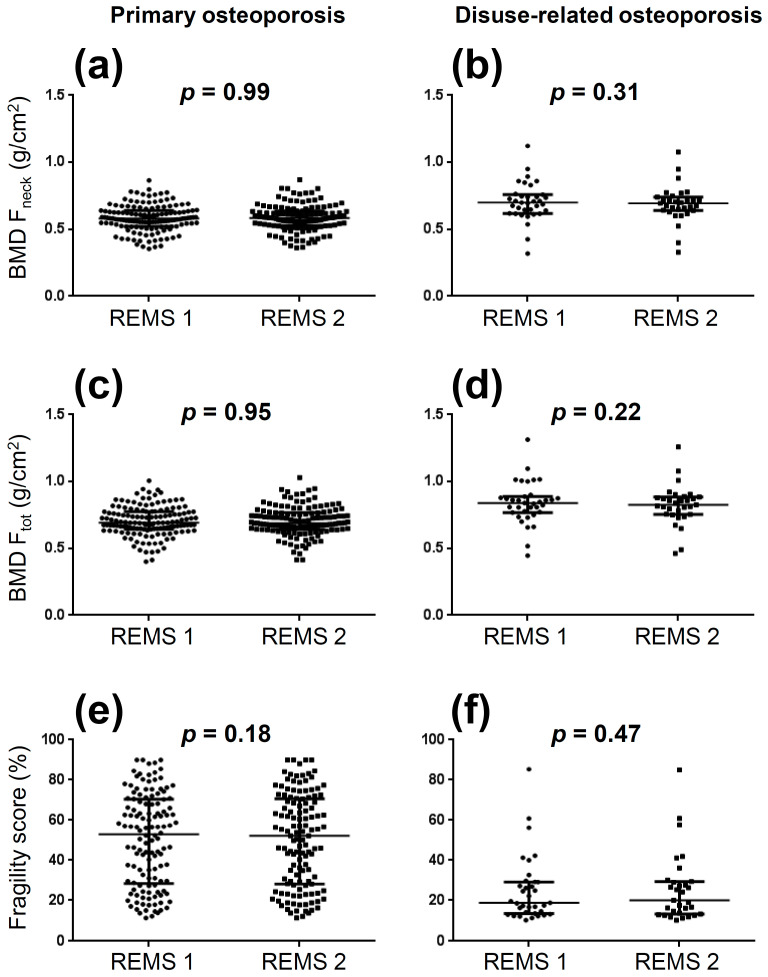
Values of femoral neck bone mineral density (BMD F_neck_: panels (**a**,**b**)), total femur bone mineral density (BMD F_tot_: panels (**c**,**d**)), and fragility score (panels (**e**,**f**)) obtained for the two REMS acquisitions (REMS 1 vs. REMS 2) in patients with primary osteoporosis (left panels) and disuse-related osteoporosis (right panels).

**Figure 2 jcm-11-03761-f002:**
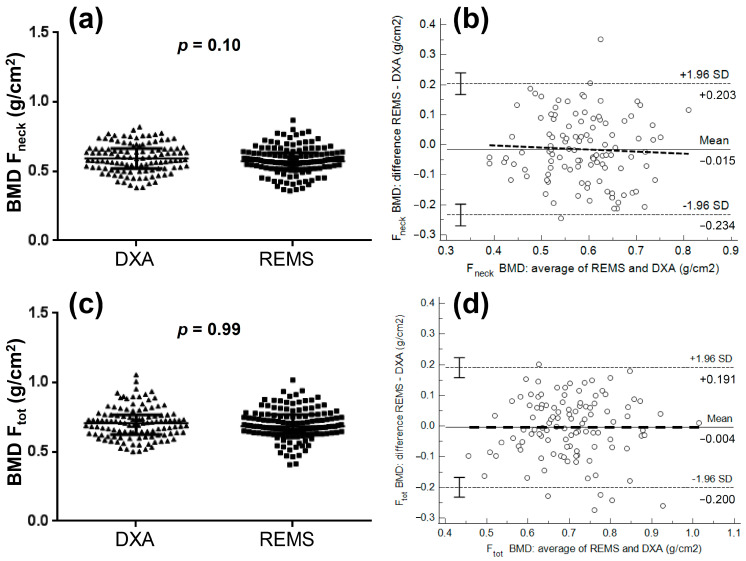
Comparisons of femoral neck bone mineral density (BMD F_neck_: panel (**a**)) and total femur bone mineral density (BMD F_tot_: panel (**c**)) between DXA and REMS and relative Bland-Altman plots (panels (**b**–**d**)) in patients with primary osteoporosis.

**Figure 3 jcm-11-03761-f003:**
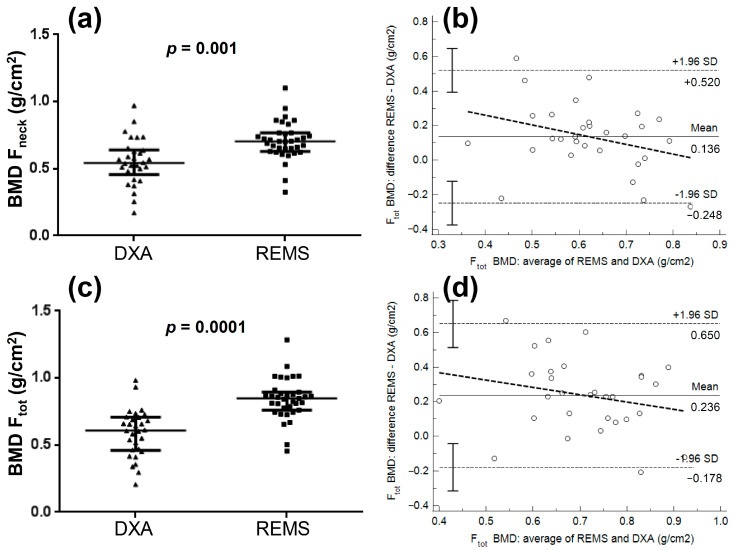
Comparisons of femoral neck bone mineral density (BMD F_neck_: panel (**a**)) and total femur bone mineral density (BMD F_tot_: panel (**c**)) between DXA and REMS and relative Bland-Altman plots (panels (**b**–**d**)) in patients with disuse-related osteoporosis.

**Figure 4 jcm-11-03761-f004:**
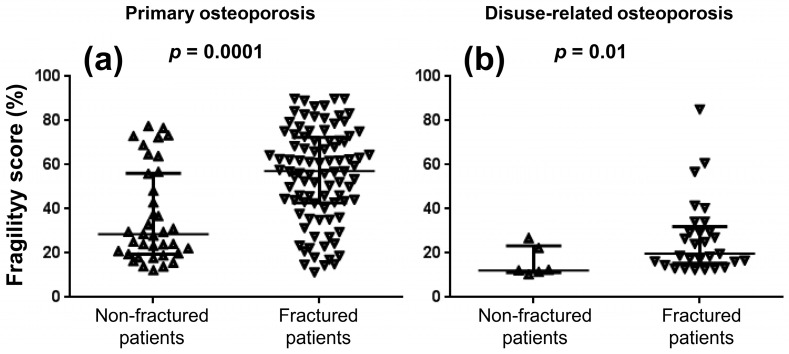
Comparisons of fragility score values between non-fractured and fractured patients with primary (panel (**a**)) and disuse-related osteoporosis (panel (**b**)).

**Table 1 jcm-11-03761-t001:** Clinical characteristics of two groups of patients. Data are reported and median (1st–3rd quartile). AIS: American spinal injury association Impairment Scale.

Variable	Primary Osteoporosis *n* = 140	Disuse-Related Osteoporosis *n* = 35
Age (years)	74.0 (64.0–81.0)	57.0 (49.0–63.5)
Gender distribution: number of females (males)	120 (20)	14 (21)
Body mass index (kg/m^2^)	23.9 (21.3–27.3)	24.7 (22.9–29.1)
Previous major osteoporotic fracture (%)	69	83
% of all patients treated with anti-osteoporotic drugs	21%	6%
% of non-fractured patients treated with anti-osteoporotic drugs	12%	0%
% of fractured patients treated with anti-osteoporotic drugs	26%	6%
FRAX score in non–complicated patients (10-yr probability of major fracture %)	10.5 (6.2–17.5)	-
FRAX score in non–complicated patients (10-yr probability of hip fracture %)	3.3 (1.8–6.4)	-
AIS score: grade A-B-C (%)	-	66-14-20
Disease history (years from the spinal cord injury)	-	15.0 (9.0–23.5)

**Table 2 jcm-11-03761-t002:** REMS reproducibility results. BMD: bone mineral density; ICC: intraclass correlation coefficient (all ICCs were statistically significant and are highlighted in bold); SEM: standard error of measurement; SDC: smallest detectable change; LSC: least significant change.

Variable	Primary Osteoporosis *n* = 135	Disuse-Related Osteoporosis *n* = 31
Femoral neck BMD		
ICC	**0.984**	**0.991**
SEM (g/cm^2^)	0.012	0.013
SDC (g/cm^2^)	0.034	0.037
LSC (g/cm^2^)	0.034	0.006
Total femur BMD		
ICC	**0.976**	**0.987**
SEM (g/cm^2^)	0.017	0.017
SDC (g/cm^2^)	0.047	0.048
LSC (g/cm^2^)	0.047	0.009
Fragility score		
ICC	**0.998**	**0.984**
SEM (%)	1.02	2.08
SDC (%)	2.84	5.76
LSC (%)	2.96	1.05

## Data Availability

The data that support the findings of this study are available from the corresponding author upon reasonable request.

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
