# Peer review of "Reproducibility and Accuracy of the Radiofrequency Echographic Multi-Spectrometry for Femoral Mineral Density Estimation and Discriminative Power of the Femoral Fragility Score in Patients with Primary and Disuse-Related Osteoporosis"

_jcm, 2022, doi:10.3390/jcm11133761_

Round 1

Reviewer 1 Report

Authors presents the research on the reproducibility and accuracy of radiofrequency echographic multi-spectrometry (REMS) for femoral BMD estimation and the reproducibility and discriminative power of the REMS-derived femoral fragility score. The main conclusions of the research are: REMS provide excellent test-retest reproducibility; the diagnostic adequacy between DXA and REMS is minimal/poor.

The article seems to be well presented, but the fourth finding in “Discussions” chapter “REMS–derived femoral fragility score discriminated between non-fractured and fractured patients in both populations of patients” seems to be excessively optimistic as (authors say about it in lines 326-330) “several non-fractured patients with primary osteoporosis had a fragility score above the median value of fragility score” and results in figure 4 do not shows clear discriminative power of the method (except statistical test). Could this discriminative power be enhanced including other parameters?

The word “probability” should be replaced by the “probably” in sentence at lines 329-330 “this subgroup of non-fractured patients could probability be considered at increased fracture risk”.

Author Response

REVIEWER

Authors presents the research on the reproducibility and accuracy of radiofrequency echographic multi-spectrometry (REMS) for femoral BMD estimation and the reproducibility and discriminative power of the REMS-derived femoral fragility score. The main conclusions of the research are: REMS provide excellent test-retest reproducibility; the diagnostic adequacy between DXA and REMS is minimal/poor.

The article seems to be well presented, but the fourth finding in “Discussions” chapter “REMS–derived femoral fragility score discriminated between non-fractured and fractured patients in both populations of patients” seems to be excessively optimistic as (authors say about it in lines 326-330) “several non-fractured patients with primary osteoporosis had a fragility score above the median value of fragility score” and results in figure 4 do not shows clear discriminative power of the method (except statistical test). Could this discriminative power be enhanced including other parameters?

ANSWER

We thank the Reviewer for the appreciation of the work and for the suggestion to introduce a note of caution in the statements relative to the discriminative power of the femoral fragility score.

In the revised manuscript, we highlighted that different subgroups (non-fractured vs fractured patents) showed differences in the fragility score and that “further studies are required to establish whether the fragility score has true discriminative power for different subgroup of patients and can therefore be considered a relevant predictor of the fracture risk”.

Moreover, the following sentence has been included:

“its discriminative power could be proved through a longitudinal study investigating the occurrence of fragility fractures in non-fractured patients presenting different baseline fragility scores”.

REVIEWER

The word “probability” should be replaced by the “probably” in sentence at lines 329-330 “this subgroup of non-fractured patients could probability be considered at increased fracture risk”.

ANSWER

The typo has been corrected.

Reviewer 2 Report

1. Please what is DXA in the abstract   2. Line 46-48: need reference   3. 61-63: need reference   4. 71-73: the originality as proposed by the authors is very clear but by searching on pubmed i saw a lot of papers on this subject, thus, the authors should explain the originality of this study   Materials and methods: 5. the number of the patients?   6. Any sample size test?   7.how can we use the two groups, one of 35 patients and the other of 140? Can we compare it?   8. Please use a, b, c.... for all the multi panels figures   9. Table 2: please mark in bold or with asterisk the significant results

Author Response

We thank the Reviewer for the appreciation of the work and for constructive comments.

REVIEWER

Please what is DXA in the abstract  

ANSWER

The meaning of the acronym has been specified.

REVIEWER

Line 46-48: need reference  

ANSWER

Two references have been added.

REVIEWER

61-63: need reference  

ANSWER

Two references have been added.

REVIEWER

71-73: the originality as proposed by the authors is very clear but by searching on pubmed i saw a lot of papers on this subject, thus, the authors should explain the originality of this study  

ANSWER

We thank the Reviewer for this comment. Although REMS is a recently developed technique, it already received considerable research attention during the last years. However, the previous studies did not investigate the research gaps we aimed to fill.

The originality of our study has been highlighted, as requested, with the addition of the following sentence:

“no previous study adopted REMS to investigate spinal cord injured patients and no previous study investigated the reproducibility and discriminative power of the femoral fragility score in osteoporotic patients

REVIEWER

Materials and methods: the number of the patients?  

ANSWER

The requested info has been added.

REVIEWER

Any sample size test?  

ANSWER

We thank the Reviewer for this comment. Yes, the sample size estimation for REMS reproducibility analysis has been performed according to the method previously proposed by Walter et al. This detail has been added, as follows:

“A sample size of at least 30 subjects (in each of the two groups) was con­sidered necessary for the test-retest reproducibility analysis, us­ing the approximate method developed by Walter et al. [22] based on α=0.05 and β=0.20, indicating an expected level of reproducibility (ρ1) of 0.98 [3] and a minimally acceptable level of reproducibility (ρ0) of 0.95”.

REVIEWER

how can we use the two groups, one of 35 patients and the other of 140? Can we compare it?

ANSWER 

We agree with the Reviewer. The two groups differed in many variables (age, gender distribution, severity of osteoporosis): therefore, no between-group comparison was performed. For all the statistical tests (with the inclusion of the REMS reproducibility assessment) the two populations were considered separately.

REVIEWER

Please use a, b, c.... for all the multi panels figures

ANSWER

All figures have been modified, as suggested.

REVIEWER 

Table 2: please mark in bold or with asterisk the significant results

ANSWER

All ICCs were statistically significant and are highlighted in bold in Table 2.

Reviewer 3 Report

The paper investigates the reproducibility and accuracy of radiofrequency Echo graphic Multi-Spectrometry (REMS) for femoral BMD estimation and the reproducibility and discriminative power of the REMS-derived femoral fragility score. REMS showed excellent test-retest reproducibility, but the diagnostic concordance between DXA and REMS was between minimal and poor. The authors may need to consider the following comments to improve the final paper:

1.      introduction section can be improved with more relevant references. 

2.      Some Operation symbols must be corrected.

3.     Table 1 must be improved. 

Author Response

REVIEWER 

The paper investigates the reproducibility and accuracy of radiofrequency Echo graphic Multi-Spectrometry (REMS) for femoral BMD estimation and the reproducibility and discriminative power of the REMS-derived femoral fragility score. REMS showed excellent test-retest reproducibility, but the diagnostic concordance between DXA and REMS was between minimal and poor.

ANSWER

We thank the Reviewer for the appreciation of the work and for constructive comments.

REVIEWER 

Introduction section can be improved with more relevant references. 

ANSWER

Several references have been added to the Introduction, as suggested by two Reviewers.

REVIEWER 

Some Operation symbols must be corrected.

ANSWER

The typos have been corrected.

REVIEWER 

Table 1 must be improved. 

ANSWER

Table 1 has been modified, as suggested.

Round 2

Reviewer 2 Report

good answers and modifications